# Speciation of Arsenic(III) and Arsenic(V) in Plant-Based Drinks

**DOI:** 10.3390/foods11101441

**Published:** 2022-05-16

**Authors:** Lena Ruzik, Małgorzata Jakubowska

**Affiliations:** Chair of Analytical Chemistry, Faculty of Chemistry, Warsaw University of Technology, 00-664 Warsaw, Poland; malgorzata.jakubowska2.stud@pw.edu.pl

**Keywords:** speciation analysis, plant-based drinks, HPLC-ICP-MS/MS

## Abstract

Recently, food products based only on plants have become increasingly popular and are often found on store shelves. It is a specific market response to the growing demand for, and interest in, plant foods. Cow’s milk has also gained its counterpart in the form of plant-based beverages, based on cereals, nuts or legumes. The emergence of an increasingly wide range of plant-based food products has also led to increased research on safe plant food consumption. This study was conducted to quantify total arsenic content and its species (arsenic(III) and (V)) in samples of plant-based beverages purchased at Polish markets. Speciation analysis of arsenic was performed by high-performance liquid chromatography combined with inductively coupled plasma mass spectrometry. The presented study was conducted on six selected plant-based beverages, including almond, millet, soybean, rice, coconut and oat. An analysis using size exclusion chromatography was performed. In order to initially visualize the content of the observed elements and the particle size of the compounds in which they occur, at first the samples were subjected to the size-exclusion chromatography. Speciation analysis of arsenic was carried out using anion-exchange liquid chromatography, combined with inductively coupled plasma mass spectrometry. The presented method was validated with certified reference material (CRM rice flour).

## 1. Introduction

In recent years, the assortment of plant-based products offered by the food industry has increased significantly [1,2,3]. Growing interest in replacing animal products in the diet is often related to allergies to ingredients of an animal origin, such as lactose or cow’s milk proteins [4]. Another reason is the change in eating habits—consumers want to eat healthy food, which often means going on a vegan diet that excludes animal products. The above factors have contributed to a significant development of the plant food industry. In addition, standard and mass-produced cow’s milk were replaced with beverages of plant origin [5,6], which most often uses ingredients such as coconut, oats, soybeans, rice, almonds, hazelnuts, walnuts and hemp seeds [7].

New plant-based products quench thirst and support the fight against stress and fatigue, increase energy, prevent ageing and even support the fight against diseases [8]. According to research results, consuming plant-based drinks helps to reduce the risk of gastrointestinal and cardiovascular diseases [9]. In addition, plant-based milk substitutes have antimicrobial properties, reduce the risk of bone disorders, support the immune system, and contain high levels of antioxidants [10]. Plant-based beverages, just like cow’s milk, can undergo fermentation processes, resulting in derived products, such as yoghurt and cheese [11,12].

Plant-based beverages belong to the group of more ecological products compared to cow’s milk [13,14]. Production of animal-origin food gives more significant cause for concern for the environment, as it requires the use of large amounts of feed and water per given unit of food product. Moreover, the area of land that needs to be used for fodder production is greater than that used for the plants that are substrates in the production of plant drinks. In addition, animal production leads to air and water pollution, overexploitation and soil degradation, and loss of biodiversity [15,16]. To sum up, the energy input per unit of fluid produced is definitely lower in the case of plant-based drinks, whose production also offers the possibility of altering the drink’s composition depending on consumer needs [17].

Plant-based beverages as a substitute for cow’s milk are made by disintegrating plant material in water. The raw material is pre-soaked, bleached, or peeled to improve treatment. Initial grinding of the raw materials provides an inhomogeneous mixture of particles of different sizes that also depends on the type of raw materials and the grinding method [18]. Homogenization is carried out to make the particle size uniform and to bring the consistency of vegetable drinks closer to that of milk. The nutritional properties of beverages are mainly influenced by the type of raw material used, its origin, the processing and enrichment process. The enrichment of plant substitutes by adding flavorings, oils, and sugars is aimed at bringing the sensory properties of the final product closer to those of cow’s milk. Ultimately, a similar chemical composition and sensory profile allow the use of plant drinks as a substitute for immediate use, and as a substrate in food production based on animal milk [19].

With the advent of new food products, the need for testing their quality and safety for humans arises [20,21]. To this end, one of the priorities is to determine the content of toxic elements most often present in the product in trace amounts, which, during long-term consumption, may accumulate in the human body, leading to severe damage or even death. The most common reason for unfavorable ingredients in plant-based products is because of their presence in the soil in which the plants are grown [22]. Typical sources of undesirable components in food are plant protection products, environmental pollution, and heavy industry in farmlands. Often, soil contamination occurs without human intervention, for example, through volcanic eruptions.

Speciation analysis is performed to identify the species of a given element in a given food product [23,24,25,26]. By a combination of several analytical techniques providing information on different properties, it is possible to determine the species in which given compounds or elements occur, even if their total content is referred to as trace amounts. This is ensured by applying hyphenated separation techniques, often chromatographic techniques, characterized by high sensitivity, with spectroscopic techniques [27,28]. Making use of the hyphenated techniques, it is possible to construct appropriate tools for speciation analysis of the various elements in plant-based drinks.

One of the food elements whose speciation analysis is best described in literature is arsenic [29,30,31,32]. It can occur in many species, from simple inorganic compounds to complex organic ones. In the simplest inorganic compounds of this element, it occurs as arsenic(III) and arsenic(V). Unconfined groundwaters can comprehend more toxic methyl derivatives of As(V) acid and less toxic derivatives of As (III) acid, arising in the biomethylation of As. In the plants, the number of species of arsenic increases because they synthesize arsenosugars, arsenolipids, arsenobetaine, or arsenomethionine [29,33,34]. These compounds are most often found in foods from the seas and oceans, such as fish, seafood, and algae, found from arsenic-rich regions. Often, arsenic compounds are also found in foods of animal and plant origin, produced in soils with high arsenic content. The tests carried out on poultry revealed the presence of 4-hydroxy-3-nitrobenzenesenoic acid, which is a feed additive showing toxic properties, in addition to the typical arsenic species [35,36]. Arsenic accumulates in the waste products of metallurgical and chemical industries and in the waste of large urban agglomerations [37]. As a result of binding by soil components, arsenic is collected in the surface layers of soil. It can also easily get from the lithosphere to the hydrosphere, making it an element often detected in natural water samples. It enters both the organisms of animals and plants [33].

The study presented was aimed at the speciation analysis of arsenic in plant-based beverages, using high-performance liquid chromatography coupled with tandem mass spectrometry (HPLC-ICP-MS/MS). Six plant drinks were analyzed: coconut, oat, almond, millet, soybean, and rice. Arsenic speciation analysis was performed on all six types of beverages using the anion-exchange liquid chromatography technique, combined with inductively coupled plasma tandem mass spectrometry (AEC-ICP-MS/MS). A simple, rapid method that requires no sample pretreatment was developed for the simultaneous determination of As(III) and As(V), using the AEC-ICP-MS/MS technique. The presented method is widely used; regardless of the origin of the plant drinks, we can use the proposed plan to conduct a speciation analysis of arsenic. However, it should be remembered that, when preparing the sample, it is necessary to consider the addition of oils during production and the addition of micro-elements that could interfere with the identified analyte.

## 2. Materials and Methods

### 2.1. Sample, Chemicals and Materials

The plant-based drinks and raw materials: oat, rice, almond, soybean, millet, and coconut-rice, were purchased from different Polish markets. A certificated reference material (IRMM-804 rice flour, Belgium) was analyzed to validate the results of the total arsenic concentrations.

The reagents used in analysis were of analytical reagent grade purchased from Sigma-Aldrich (Sigma-Aldrich, Buchs, Switzerland). The nitric acid of purity for trace metal analysis was purchased from Fluka (Buchs, Switzerland). Ultra-pure water (15 MΩ cm) was obtained with Milli-Q Elix 3 Water Purification system Millipore (Molsheim, France). 

The SEC column was calibrated, using the size exclusion standard (BIO-RAD, Warsaw, Poland). The calibration curves were prepared using a solution of Environmental Spike Mix (1000 mg L^−1^ of Fe, K, Ca, Na, Mg and 100 mg L−1 of Ag, Al, As, Ba, Be, Cd, Co, Cr, Cu, Mn, Mo, Ni, Pb, Sb, Se, Tl, V, Zn, U; matrix 5% HNO3) purchased from Agilent Technologies.

### 2.2. Instrumentation

Samples of extracts were injected on a Superdex200 10/300GL (GE Healthcare Life Sciences, Chicago, IL, USA) column coupled to the ICP-MS/MS instrument by directly connecting the column outlet to the cross-flow nebuliser through a PEEK tubing. Chromatographic separations were performed using Agilent 1260 Infinity II LC System gradient HPLC pump (Agilent Technologies, Tokyo, Japan). The operational parameters of HPLC are summarized in Table 1.

An Agilent 8900 ICP Triple Quadrupole Mass Spectrometer (Tokyo, Japan) was used as an element-specific detector for quantifying metal content in plant-based drinks. The spectrometer interference was equipped with Pt cones. The position of the torch and the nebuliser gas flow were adjusted daily, emphasizing the decrease of the level of CeO^+^ (below 0.2%) to minimize the risk of occurrence of polyatomic interference caused by oxides. The RF power was 1550 W, nebuliser gas flow—0.95 L min^−1^, and optional gas flow (oxygen)—0.375 mL min^−1^ and monitored isotopes: ^75^As. The working conditions were optimized daily using a 10 µg L^−1^ solution of ^7^Li^+^, ^89^Y^+^ and ^209^Bi^+^ in 2% (*v*/*v*) HNO_3_. Analyses were performed in the Time-Resolved Analysis (fast TRA) mode, using a dwell time of 0.1 ms (100 μs) per point with no settling time between measurements.

A Bandelin Sonorex Model 1210 ultrasonic bath (Germany), MPW Model 350R centrifuge (MPW Warsaw, Poland), and water bath with thermostatically controlled temperature (Mammert, Germany) were used during mixing procedures. Microwave digestion Speedwave^®^ four Berghof (Germany) was used for mineralization samples and during the extraction procedure.

### 2.3. Sample Preparation

The mineralization of samples toward arsenic determination in plant-based drinks: The plant-based beverages (5 mL) and rice flour were digested by microwave-assisted mineralization with a mixture of 5 mL of HNO_3_ and 3 mL of H_2_O_2_. The digests were diluted to a final volume of 25 mL with Milli-Q water. Further dilutions for ICP-MS/MS analysis were prepared using 2% (*v/v*) nitric acid solution and 10 ng mL^−1^ of yttrium (^89^Y) as an internal standard.

**Speciation analysis.** The solutions of the arsenic standard were prepared: the first contained 0.0517 g of NaAsO_2_ in 30 mL of water, and the second contained 0.125 g of Na_2_HAsO_4_ in the same volume of deionized water. Then 5 mL of both solutions were added, thus creating a mix. Subsequently, standards solution and plant-based drinks were filtered with a 0.2 µm pore size filter, and then for beverage samples, the 10 kDa centrifugal filters were used. The samples prepared in this way were analyzed using strong anion-exchange liquid chromatography combined with tandem mass spectrometry with inductively coupled plasma (AEC-ICP-MS/MS).

## 3. Results

### 3.1. The Total Content of Arsenic in Plant-Based Drinks

The total concentration of elements in plant-based drinks was measured by ICP-MS/MS. Results represent the average amount established for three samples (each measured three times) and show the total concentration of elements (Table 2). The concentrations of arsenic in drinks depend strongly on the type of plant used for the production of beverages its the highest concentrations were found in the beverages based on rice and coco-rice, which confirms the high concentration of this element in rice. Similar, but lower than that in rice were the concentrations of arsenic in the samples of beverages based on millet, soya and oat. 

To indicate the expected value of maximum arsenic contents in beverages, the content of those elements in the dry and soaked materials was carried out. This process was supposed to extract arsenic from the raw materials, similar to the production of plant-based drinks. Different concentrations of arsenic were found in raw products used to produce the beverages and in water left after soaking natural materials: soya, rice, coconut, and oat (for 24 h in a dark place) (Table 2). It is important to mention that the highest concentration of arsenic was observed in the water left after 24 h soaking of soybean, rice, coconut, and oat. The lowest arsenic concentrations were observed for millet and almond. 

The lowest concentration of arsenic was determined in the beverages based on almonds; thus, they seem to be the safest for consumption (Figure 1). The next step was the speciation analysis of arsenic to identify the species presented in plant-based beverages.

#### Method Validation

The presented method was validated using certified reference material (CRM) rice flour. The same material was applied to optimize the analytical method for arsenic speciation analysis. The total content of the elements was determined. Table 3 presents the results of the concentration calculations obtained assuming the CRM material mineralization process and the ICP-MS/MS analysis results.

Based on the obtained results, it can be concluded that the experimentally obtained metal contents in the CRM are of the same order of magnitude as listed by the manufacturer, and mainly within the given values of errors.

The applied method of elemental analysis was validated following the international guidelines described in the ISO/IEC 17025:2005 protocol, based on a determination of linearity, precision, accuracy, and limits of detection (LOD) and quantification (LOQ).

The method’s precision was evaluated by analyzing the ten independent experiment preparations for each metal—the test samples against the internal standard and the %RSD of metals calculated. The accuracy of the obtained data was high and repeatable (%RSD), and was in the range of 0.68–3.62%.

The linearity test was performed to check the capacity of the entire analytical system to display a linear response and proportionality of the signal intensity to the relevant concentration of the analyte varied within a certain range. The obtained results’ dependence on the analyte concentration were linear, in the range 0.5 µg L^−1^–100.0 µg L^−1^, with r^2^ above 0.999. The linear regression data for the calibration plot suggest a good linear relationship between the intensity of the signal assigned to the metal contained in the analyte and its concentration over a wide range. Column recoveries that were greater than 95% were obtained for arsenic speciation analysis in plant-based beverages.

LOD, based on the values of standard deviations (SD) from 10 measurements of blank, was calculated as 0.8–1.2 µg kg^−1^, while LOQ (3.3 LOD) was estimated as 2.6–4.0 μg kg^−1^.

### 3.2. SEC-ICP-MS Characteristics of Various Extracts Containing Arsenic Species

The extracts containing arsenic species were characterized by SEC-ICP-MS/MS analysis in the first step. (Figure 2). Size exclusion chromatography in conjunction with to inductively coupled plasma mass spectrometry allows the separation and element-specific detection of the eluted species.

Two signals were recorded upon the separation of arsenic compounds present in the millet-based beverages. The first observed signal had an intensity of about 4000 cps and the second one—3000 cps, both of which were 44–1.35 kDa. Analysis of the coconut-based beverages revealed the presence of one signal of intensity of about 5000 cps in the 17–1.35 kDa range. Upon separating the arsenic compounds in the oat-based beverages, two signals with intensities of about 1000 cps were observed, in the range of 44–1.35 kDa. Two signals with intensities of about 6000 cps in the particle size range of 44–1.35 kDa were found for the rice-based beverages. The analogous separation analysis for the arsenic soybean beverages produced two signals with intensities of only 1000 cps, in the particle size range of 44–1.35 kDa. For the almond-based drinks, no significant signals assigned to arsenic were observed.

The SEC separation was used for the fractionation of the plant-based drinks to characterize the arsenic compounds. The results obtained from the SEC-ICP-MS investigation indicate that only two fractions of arsenic compounds were observed in plant-based beverages—the medium and small particle size arsenic compounds. Only small particle size arsenic compounds are present in the plant-based drinks, and we can use the AEC chromatography to identify them.

### 3.3. AEC-ICP-MS Characteristics of Various Extracts Containing Arsenic Species

Speciation analysis of arsenic compounds in various plant-based beverages was performed, based on AEC-ICP-MS chromatograms of the samples studied, shown in Figure 3. AEC-ICP-MS/MS chromatograms were also recorded for the standards of arsenic(II) and (V) and are presented in Figure 3c. The signal in the chromatogram corresponding to the retention time of about two minutes comes from the compounds containing AsO^2−^ ions. The second signal corresponding to the retention time of 7 min comes from the compounds containing AsO_4_^3−^ ions.

As follows from analysis of the obtained results, the most significant amount of total arsenic compounds containing both analyzed species was observed in the rice-based beverages (Figure 3b). Significant amounts of the two arsenic species were also detected in the coconut-rice-based and millet-based beverages (the green and blue chromatograms in Figure 3a). In addition, the chromatograms obtained for the soya-based and almond-based beverages confirmed the presence of both arsenic species, but in small amounts. In contrast, the smallest number of arsenic compounds was detected in the oat-based beverages (for these samples, no chromatograms are shown, as they are uninformative).

It is important to mention that we could not observe, on the AEC-ICP-MS/MS chromatograms for all of the investigated plant-based beverages, the signals for the organic forms of arsenic: monomethylarsonic acid (MMA(V)); dimethylarsinic acid (DMA(V)); arsenocholine (AsC); or arsenobetaine (AsB). It could indicate that the mentioned species are not present in plant-based beverages.

## 4. Conclusions

Speciation analysis of arsenic in plant-based beverages was performed. The first step of the study was the fractionation of arsenic compounds present in six plant drinks: almond, coconut-rice, millet, soybean, oat, and rice, using size-exclusion liquid chromatography (SEC) combined with inductively coupled plasma mass spectrometry (ICP-MS/MS). As a result of the separation, the arsenic fractions present in the tested samples and the particle size distribution of the compounds were obtained. As a result of the conducted analyses, arsenic compounds containing AsO^2−^ and AsO4^3−^ ions to a greater or lesser extent were observed in each of the six tested plant drinks. It should be mentioned that inorganic arsenic induces a variety of toxicities, including cancer. The two forms of inorganic arsenic, arsenate (AsV) and arsenite (AsIII), are easily taken up by the plant, and As(V) can be readily converted to As(III), which is the more toxic of the two mentioned forms.

According to the results of our study, the average content of arsenic in plant-based beverages varied in the range 0.02 to 2.34 μg L^−1^. Our estimation of the daily intake of inorganic arsenic from plant-based drinks consumption indicated that it was higher than the BMDL_0.1_ value (0.3 μg/kg BW/day) set by the European Food Safety Authority (EFSA) for every age group [38]. Oat and almond-based beverages have been found the safest for consumption.

## Figures and Tables

**Figure 1 foods-11-01441-f001:**
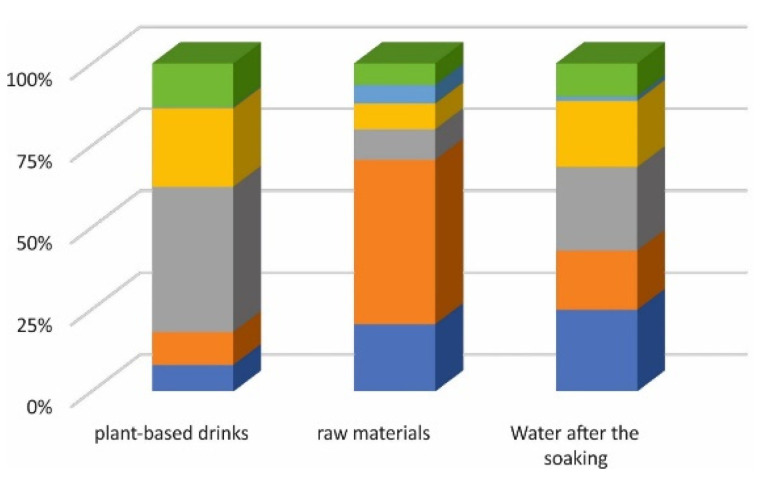
The relationship of the total content of arsenic in plant-based drinks, raw material and water after soaking: blue—soy; orange—oat; grey—rice; yellow—coconut-rice; light blue—almond; green—millet.

**Figure 2 foods-11-01441-f002:**
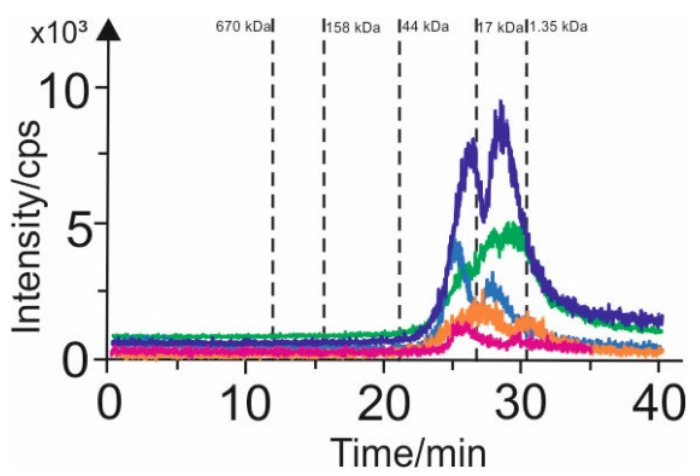
SEC-ICP-MS chromatograms of arsenic (75As), obtained for plant-based beverages: pink line—soya-based, orange line—oat-based, blue line—millet-based, green line—coconut-based, violet line—rice-based.

**Figure 3 foods-11-01441-f003:**
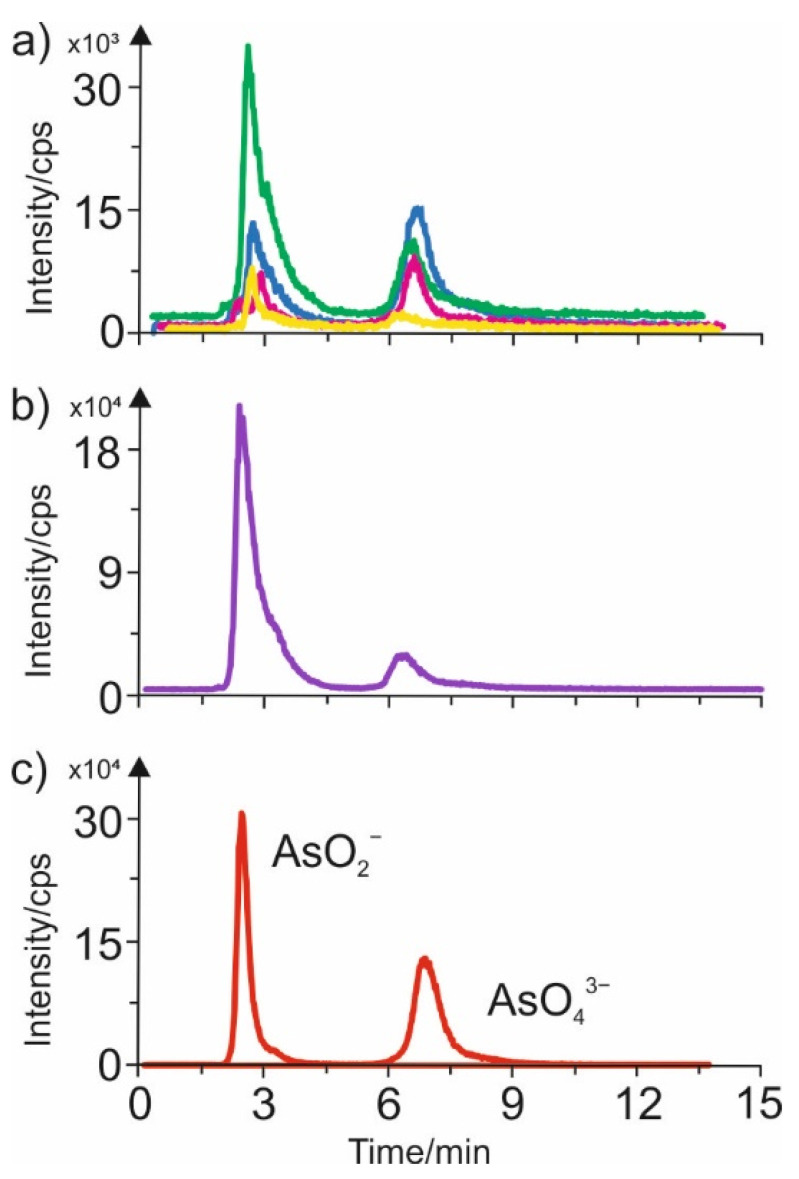
AEC-ICP-MS/MS chromatograms obtained for plant-based beverages: (**a**) ^75^As isotope for coconut-based (green line), millet-based (blue line), soya-based (pink line), almond-based (yellow line); (**b**) rice-based (violet line) and (**c**) standard mixture (red line).

**Table 1 foods-11-01441-t001:** Operational parameters for HPLC.

Settings	
Pump	Agilent 1260 Series
Column	Superdex 200 (10 × 300 mm × 10 µm)—GE Healthcare Life Sciences
Mobile phase	10 mM ammonium acetate buffer (pH 7.4)
Elution program	isocratic
Flow	0.75 mL min^−1^
Injection volume	100 µL
Column temperature	28 °C
Pump	Agilent 1260 Series
Column	Agilent io SAX (5 µm, 4.6 × 250 mm)
Mobile phase	5 mM sodium dihydrogen phosphate (pH 6.2) and 0.2 mM EDTA
Elution program	isocratic
Injection volume	100 µL
Flow rate	0.7 mL min^−1^
Column temperature	22 °C

Calibration of SEC column was made with a standard mixture of thyroglobulin (670 kDa) tr = 11.8 min; γ-globulin (158 kDa) tr = 14.8 min; ovalbumin (44 kDa) tr = 21.4 min; myoglobin (17 kDa) tr = 27.3 min; and vitamin B12 (1.35 kDa) tr = 30.2 min.

**Table 2 foods-11-01441-t002:** The total content of arsenic in plant-based drinks, raw material and water after the soaking.

	Total Content of Arsenic					
	(μg L^−1^)	RSD (%)	(μg g^−1^)	RSD (%)	(μg L^−1^)	RSD (%)
	Type of Beverages		Raw Material		Water after the Soaking	
Soybean	0.41970	1.06	0.03412	2.25	0.33071	2.83
Oat	0.53216	2.29	0.08410	2.23	0.24138	2.52
Rice	2.34086	2.70	0.01566	1.47	0.34022	3.13
Coconut-rice	1.26839	3.62	0.01311	1.89	0.26989	2.42
Almond	0.01616	1.95	0.00941	2.44	0.01784	0.68
Millet	0.70325	2.23	0.01110	1.45	0.13494	1.95

**Table 3 foods-11-01441-t003:** The average total content of the arsenic in rice flour, CRM (*n* = 3, ±SE).

Total Content of Arsenic					
(mg/kg)	After Mineralization (μg kg^−1^)	Dilution ×5 (μg kg^−1^)	RSD (%)	Dilution ×30 (μg kg^−1^)	RSD (%)
calculations					
0.0490 ± 0.0011	0.9801 ± 0.0137	0.2023 ± 0.0571	2.34	0.0301 ± 0.0020	3.11
experimentals					
-	-	0.1810 ± 0.0350	3.05	0.0206 ± 0.0012	2.89

## Data Availability

The datasets generated for this study are available on request to the corresponding author.

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
