# Peer review of "Speciation of Arsenic(III) and Arsenic(V) in Plant-Based Drinks"

_foods, 2022, doi:10.3390/foods11101441_

Round 1
Reviewer 1 Report
The present manuscript was conducted to quantify total arsenic content and its species arsenic (III) and (V)) in samples of plant-based beverages purchased at Polish markets. Speciation analysis of arsenic was performed by high-performance liquid chromatography combined with inductively coupled plasma mass spectrometry.
The manuscript presents the hazard of arsenic content in the field of plant-based food and then detection by high-performance liquid chromatography combined with inductively coupled plasma mass spectrometry, which generally meets the journal scope. The manuscript can make contribution to the research carried out by the groups that are dedicated to this particular topic. Consider please the following comments.
Introduction
- The authors hereby introduced the toxic elements in thesix selected plant-based beverages and further tried to display the developments of the effective materials and methods of detection, pointing out that this manuscript is aimed at the comprehensive and critical application of arsenic in the detection of pesticides from plant-based products.
However, the introduction of the manuscript is prolix, which should be highlighted the key points of arsenic valence state, other content for auxiliary explanations.
- The logic beneath development and advantage of this detection method is not clear enough.The authors should answer this question through an in-depth explanation.
- Note important information in the Figure 1, such as the content value.
- There are formatting problems in the Table, for example, three wire table.
- Figure 3 lacks the legend of b and c, which need to be checked carefully.
- The content comparison of AsO2- and AsO43-should be added in the conclusion, and the influence of valence state difference should be discussed.
Author Response
Reviewer #1 comments:
The present manuscript was conducted to quantify total arsenic content and its species arsenic (III) and (V)) in samples of plant-based beverages purchased at Polish markets. Speciation analysis of arsenic was performed by high-performance liquid chromatography combined with inductively coupled plasma mass spectrometry.
The manuscript presents the hazard of arsenic content in the field of plant-based food and then detection by high-performance liquid chromatography combined with inductively coupled plasma mass spectrometry, which generally meets the journal scope. The manuscript can make contribution to the research carried out by the groups that are dedicated to this particular topic. Consider please the following comments.
Introduction - The authors hereby introduced the toxic elements in thesix selected plant-based beverages and further tried to display the developments of the effective materials and methods of detection, pointing out that this manuscript is aimed at the comprehensive and critical application of arsenic in the detection of pesticides from plant-based products. However, the introduction of the manuscript is prolix, which should be highlighted the key points of arsenic valence state, other content for auxiliary explanations.
In the introduction from line 83 to line 92, we tried to outline the most important information regarding the arsenic species and their impact on the organisms, giving references from 29 to 34 in which these properties are widely described, because of the communication format. However, information on the identified forms in plant-based drinks has been added to the conclusions.
The logic beneath development and advantage of this detection method is not clear enough. The authors should answer this question through an in-depth explanation.
The important information has been added to the Introduction paragraph.
Note important information in the Figure 1, such as the content value.
The content value are presented in Table 2, we don't want to duplicate the same information
There are formatting problems in the Table, for example, three wire table.
It was checked.
Figure 3 lacks the legend of b and c, which need to be checked carefully.
It has been added.
The content comparison of AsO2- and AsO43-should be added in the conclusion, and the influence of valence state difference should be discussed.
In our method, we didn’t make the quantitative analysis of arsenic species, only the identification of the presented form. The information about the different toxicity of the arsenic (III) and (V) has been added.
Reviewer 2 Report
The paper described the speciation of arsenic(III) and (V) in six different plant-based drinks. This topic is very interesting due to the increasing consumption of these beverages. However in the present form lacks of a logical presentation of the data.
The title is referred to plant-based drink, but the total content of arsenic was also performed on raw material and on water after the soaking. What is the significance of these determinations? Why did the authors performed them? In addition, the content is expressed in microgram per liter....so, did the author perform an extraction on raw material? Nothing is reported about this.
Section 3.1.1 Validation of CRM analysis? What is it? Better explain.
An important key point is the fact that it is not clear the logical sequence of analyses. Why did the authors performed SEC anlayses? Did they perform AEC analyses of the fraction isolated using SEC? It is not clear
In addition, in Materials and methods section, the instrumentation part is confuse. In table 1 the parameters for AEC analysis are reported, but in the text are lacking.
The Authors indicated the two fractions in SEC profiles as medium and small particle size arsenic compounds, but it is very difficult to define the detected arsenic species in this way!!!
Line 244: ..."the most significant amount..": how did the authors perform the quantification? The method validation reported in section 3.1.1 is referred to this quantification?...it is not clear
Minor concerns: deep English revision, many typing errors are present and some sentences are meaningless.
Author Response
Reviewer #2 comments:
The paper described the speciation of arsenic(III) and (V) in six different plant-based drinks. This topic is very interesting due to the increasing consumption of these beverages. However in the present form lacks of a logical presentation of the data.
The title is referred to plant-based drink, but the total content of arsenic was also performed on raw material and on water after the soaking. What is the significance of these determinations? Why did the authors performed them? In addition, the content is expressed in microgram per liter....so, did the author perform an extraction on raw material? Nothing is reported about this.
To indicate what maximum arsenic contents can be expected in beverages, tests of the content of the element in dry and soaked material were carried out. This process was supposed to extract arsenic from the raw materials in a similar way to the production of plant-based drinks. Thank you very much for this remark: the content of arsenic was measured in raw materials, and should be expressed in µg g-1.
Section 3.1.1 Validation of CRM analysis? What is it? Better explain.
The title of this subsection has been changed for Method validation. The unfortunate shortcut has been used.
An important key point is the fact that it is not clear the logical sequence of analyses. Why did the authors performed SEC anlayses? Did they perform AEC analyses of the fraction isolated using SEC? It is not clear.
The SEC separation was used for fractionation of the plant-based drinks, to characterize the arsenic compounds. This part of our investigations confirmed that only small particle size arsenic compounds are present in the plant-based drinks and we can use the AEC chromatography to the identification.
In addition, in Materials and methods section, the instrumentation part is confuse. In table 1 the parameters for AEC analysis are reported, but in the text are lacking.
The authors concluded that the information contained in the tables should not be duplicated in the text, hence the parameters of the work are presented only in the form of a table.
The Authors indicated the two fractions in SEC profiles as medium and small particle size arsenic compounds, but it is very difficult to define the detected arsenic species in this way!!!
It was not our goal to perform arsenic speciation analysis using the SEC. During the investigation, we only fractionate the arsenic compounds, and for the identification of arsenic species was used the AEC chromatography and the standard compounds.
Line 244: ..."the most significant amount..": how did the authors perform the quantification? The method validation reported in section 3.1.1 is referred to this quantification?...it is not clear
This information relates to the highest total arsenic content of the rice drink. And such results have been marked and presented in Table 2.
Minor concerns: deep English revision, many typing errors are present and some sentences are meaningless.
The correction proof has been made before the submission of the manuscript.
Round 2
Reviewer 2 Report
The Authors only partially revised the paper according the suggestions.
They answered to each Reviewer's commenì, but in the text they did not report all the explanation. The reader of the Results section can find not so easy to better understand step by step the experimental work. I suggest to improves this.
Author Response
Reviewer #2 comments:
The Authors only partially revised the paper according the suggestions.
They answered to each Reviewer's commenì, but in the text they did not report all the explanation.
We did not intend to fail to respond to the Reviewer's comments. We tried to respond to all comments, and the text introduced the changes. Not all comments agreed, hence perhaps the Reviewer's impression that he did not see amendments in the text.
Here is the answer to the Reviewer's comment, along with the relevant information included in the text of the manuscript
The reader of the Results section can find not so easy to better understand step by step the experimental work. I suggest to improves this.
Thank you very much for these comments. The Results section has been improved, and we hope now will be better understood by Readers.